# Evaluating the Path to the European Commission's Organic Agriculture Goal: A Multivariate Analysis of Changes in EU Countries (2004–2021) and Socio-Economic Relationships

**Stefan Krajewski [1], Jan Žukovskis [2] , Dariusz Gozdowski [3] , Marek Cieśliński [1] and Elżbieta Wójcik-Gront [3],***

[1]  Ministry of Agriculture and Rural Development, Wspólna 30, 00-930 Warsaw, Poland;
    stefan.krajewski@minrol.gov.pl (S.K.); marek.cieslinski@minrol.gov.pl (M.C.)
[2]  Department of Business and Rural Development Management, Vytautas Magnus University,
    53361 Kaunas, Lithuania; jan.zukovskis@vdu.lt
[3]  Department of Biometry, Institute of Agriculture, Warsaw University of Life Sciences, Nowoursynowska 159,
    02-776 Warsaw, Poland; dariusz_gozdowski@sggw.edu.pl
*  Correspondence: elzbieta_wojcik_gront@sggw.edu.pl; Tel.: +48-225-932-645

**Abstract:** This study comprehensively analyzed the dynamic landscape of organic farming in the European Union (EU) from 2004 to 2021, investigating the shifts in dedicated agricultural areas influenced by evolving preferences and the priorities of farmers and consumers. Examining the impact of socio-economic factors, including gross domestic product (GDP) per capita, the human development index (HDI), and human population density, this study established multivariate relationships through country-level analyses based on correlations, principal component analysis, cluster analysis, and panel analysis. Despite a universal increase in the organic agriculture areas across all the EU countries during the study period, the production levels exhibited negative correlations with the human population density, GDP per capita, and HDI. Notably, the Baltic countries and Austria led in organic farming production, while Malta, the Netherlands, Belgium, Ireland, and Luxemburg formed a distinct group in the cluster analysis with less intensive organic agriculture per capita. These insights are crucial for supporting the resilience and sustainability of organic farming as it continues to evolve. Predictions of organic agriculture share for 2030 based on trends evaluated using linear regression in the years 2004–2021 estimated about 12% of utilized agricultural area, which was much lower than the target share of the European Commission at 25%. Predictions based on linear regression showed that achieving the European Green Deal target of a 25% share of organic agriculture in unlikely in most EU countries by 2030. The target is only highly probable to be obtained in Austria, Estonia, and Sweden. The EU countries varied significantly across various indices characterizing organic agriculture, including organic agriculture area share. It should be noted that the study was conducted on the data obtained prior to the outbreak of the war in Ukraine, which could potentially alter the previous trends in the development of organic agriculture in the EU.

**Keywords:** Common Agricultural Policy (CAP); Green Deal; multivariate analysis; long-term changes; organic agriculture

## 1. Introduction

The transformation of agriculture in the European Union (EU) has been underscored by an increasing recognition of the environmental and societal challenges linked to conventional farming methods [1,2]. Organic farming, as a sustainable alternative, has garnered substantial attention in recent decades [3]. This shift toward organic agriculture aligns with global initiatives addressing critical issues such as climate change, biodiversity depletion, and food security. In pursuing ecologically intensive farming systems, it is important to balance environmental sustainability with production reliability. It is possible, especially in cases where there is a presence of livestock and a diversification of crops [4]. Organic farms

have consistently excelled in environmental metrics; however, they exhibit greater yield variability compared to their conventional counterparts [5], which offer higher production yields with reduced variability [6]. Sustainable agriculture necessitates the provision of nutritious food, the minimization of environmental impacts, and the assurance of fair incomes for producers. Achieving these ambitious goals requires incremental enhancements in conventional agricultural practices and a transformative shift toward agroecological principles [7]. The realm of agricultural management holds potential in the sequestration of carbon and the mitigation of climate change. The transition from conventional tillage to no-till practices has proven effective in sequestering significant carbon. Notably, the rates of carbon sequestration vary, with the shift from conventional tillage to no-till reaching equilibrium within 15–20 years, while enhanced rotation complexity may take 40–60 years to stabilize [8]. These findings underscore the importance of sustainable agricultural practices in combating climate change and promoting long-term soil health. The regulatory framework for organic agriculture in the EU is defined by established regulations, including European Union Organic Farming Regulation (EC) No 834/2007, which emphasizes the avoidance of synthetic pesticides, chemical fertilizers, and genetically modified organisms (GMOs). Organic farming fosters sustainable soil and water management, biodiversity preservation, the protection of natural ecosystems, and the assurance of high animal welfare standards, with particular attention to outdoor access and organic feed.

To attain organic certification within the EU, farms and agricultural products undergo rigorous evaluation by accredited organic certification bodies. Certification ensures adherence to stringent organic farming standards, enabling certified products to bear the EU organic logo and the "organic" label. National competent authorities in EU member states carry out oversight and enforcement of organic agriculture regulations. Furthermore, the EU provides vital support and funding for organic agriculture through the Common Agricultural Policy (CAP) [9] and rural development programs, offering financial incentives to encourage the adoption and maintenance of organic farming practices. In recent years, the EU member states supported organic farming in almost 5% of EU-utilized agricultural area (UAA), at an average cost of about 200 euros per ha [10]. The area supported represented about 2/3 of the total certified organic land area in the EU. Support was also provided in other forms, e.g., education, research, consumer promotion, and the development and implementation of EU regulations defining organic food and farming. The organic market in the EU has been steadily expanding, driven by growing consumer demand for healthier and more sustainable food choices.

The European Green Deal and the Farm to Fork Strategy are integral parts of the EU's efforts to achieve climate neutrality and sustainable food systems. These strategies are closely linked to biodiversity conservation, aiming to protect ecosystems and halt biodiversity loss. Organic farming plays a vital role in supporting these goals by promoting practices that enhance biodiversity and prioritize soil health. The EU Biodiversity Strategy for 2030 emphasizes protected areas and nature restoration targets. By embracing organic practices, the EU can move closer to its vision of sustainable and resilient food systems. The European Commission has recently unveiled an ambitious initiative, setting a target to elevate the share of organic agriculture to 25% within the continent by 2030 [11]. The member states have committed close to 3 billion euros to supporting organic farming per year in the five-year period 2023–2027, which represents a 50% increase in comparison to previous years [10]. This may be sufficient to deliver an increase in organic land area to 15% by 2027 and to deliver the 25% target by 2030, which is part of the Green Deal, Farm to Fork Strategy, and the Biodiversity Strategy. This target carries substantial implications, impacting not only the agricultural sector but also the environment, economy, and the well-being of European citizens. The expansion of organic agriculture promises reduced soil degradation, diminished water pollution, and a positive contribution to climate change mitigation through carbon sequestration. Additionally, the shift toward organic farming often involves localized, smaller-scale production, fostering economic resilience in rural communities and stimulating regional development.

The European Commission's ambitious initiative to elevate the share of organic agriculture to 25% by 2030 aligns with the growing consumer demand for organic products [12]. To assess the feasibility of this initiative, it is crucial to consider factors such as farmers' ability to transition to organic practices, potential impacts on food prices, and the necessary support and infrastructure for organic farmers. Challenges, including potential decreases in crop yields during the conversion period, must also be addressed [13].

Promoting organic farming and increasing the share of organic agriculture requires a substantial boost in consumer demand for organic products [14]. Thus, there is a need to raise awareness about the benefits of organic farming and build consumer trust through effective marketing campaigns, educational initiatives, and transparent labeling systems [15].

The study evaluated whether the development of organic farming was correlated with socioeconomic indicators such as gross domestic product (GDP) per capita, the human development index (HDI), human population density, as well as the share of employment in agriculture at the country level within the EU. These socioeconomic indicators are commonly used in various studies and provide a comprehensive characterization of the economic status of countries. We hypothesized that they may be associated with the development of organic farming and could help explain the variability in organic farming development among EU countries.

Therefore, this study aimed to comprehensively examine the dynamic landscape of organic farming, emphasizing changes in the area dedicated to organic agriculture within EU countries from 2004 to 2021. Moreover, predictions for the organic agriculture area in 2030 were presented to evaluate the feasibility of achieving the European Green Deal target of 25% organic farming in the EU by that year. Furthermore, the study sought to explore and identify the socio-economic variables that may have influenced observed trends in organic farming, establishing multivariate relationships at the country level. The years 2004–2021, marked by expansions, policy reforms, and economic fluctuations, were crucial for gaining insights that could inform the development of more effective policies and strategies. These insights are essential for supporting sustainable agricultural practices and addressing socio-economic challenges prevalent in the agricultural sector.

## 2. Materials and Methods

### 2.1. Sources of Data and Description of Variables

Data sourced from the Food and Agriculture Organization Corporate Statistical Database (FAOSTAT) [16], Forschungsinstitut für Biologischen Landbau (FiBL) [17], Eurostat [18] and the World Bank [19], covering the time frame from 2004 to 2021, were employed to investigate changes in the agriculture area under organic agriculture. The study concentrated on a set of 27 EU countries (Figure 1). The statistical analyses were performed using 17 variables that characterized the status of organic farming production and socio-economic conditions in the studied countries. Abbreviations and descriptions of the variables are presented in Table 1.

**Table 1.** Variables that characterize organic agriculture and socio-economic conditions were included in the study.

| Abbreviation | Description of the Variable | Units |
|---|---|---|
| OAS | Organic area share of utilized agricultural area | % |
| GDPC | Gross domestic product (GDP) per capita | USD |
| HDI | Human development index | - |
| EAS | Share employed in agriculture | % |
| HP/A | Human population/area of total country | Thousand people/thousand ha |
| OP/UAA | Organic producers/utilized agricultural area | Producers/M ha |
| CR/UAA | Cereals, organic area/utilized agricultural area | ha/M ha |
| PP/UAA | Pulses and protein crops, organic area/utilized agricultural area | ha/M ha |

**Table 1.** *Cont.*

| Abbreviation | Description of the Variable | Units |
|---|---|---|
| FR/UAA | Fruits total, organic area/utilized agricultural area | ha/M ha |
| GR/UAA | Grapes, organic area/utilized agricultural area | ha/M ha |
| OL/UAA | Oilseeds and olives, organic area/utilized agricultural area | ha/M ha |
| RC/UAA | Root crops, organic area/utilized agricultural area | ha/M ha |
| VG/UAA | Vegetables, organic area/utilized agricultural area | ha/M ha |
| BV/UAA | Organic bovine animals/utilized agricultural area (M ha) | animals/M ha |
| PG/UAA | Organic pigs/utilized agricultural area | animals/M ha |
| PO/UAA | Organic poultry/utilized agricultural area | animals/M ha |
| SH/UAA | Organic sheep/utilized agricultural area | animals/M ha |

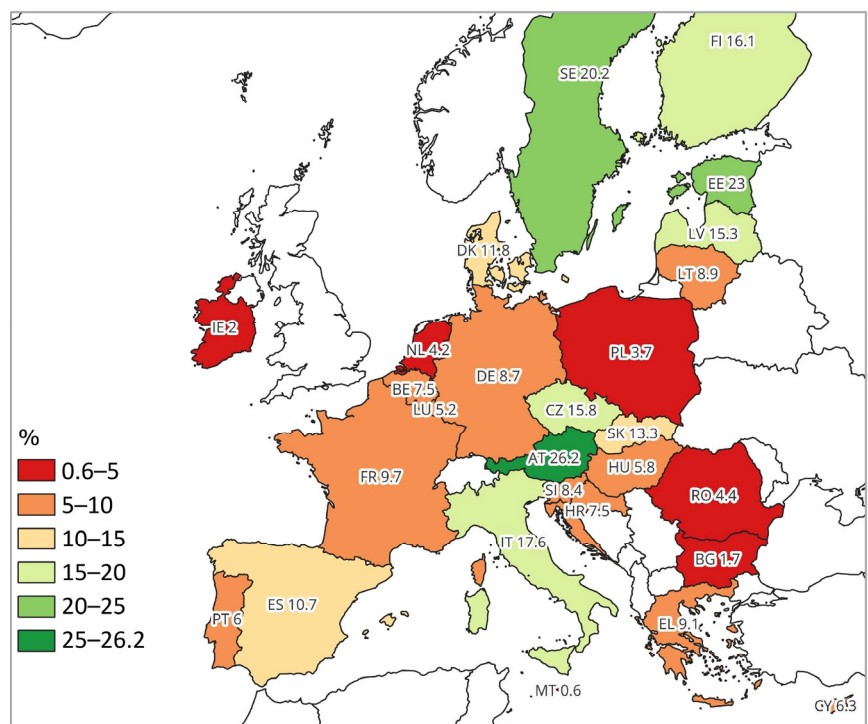

**Figure 1.** Countries included in the analysis and percentage of organic agriculture as a share of the total utilized agricultural area (UAA) for 2021.

### 2.2. Statistical Analysis

Linear regression analysis was employed to investigate the relationships between time and changes in the area dedicated to organic agriculture in the EU countries [20]. Panel analysis based on a linear model of multiple regression [21] was employed for the evaluation of long-term changes in the percentage of organic agriculture areas. The independent variables were gross domestic product (GDP) based on purchasing power parity (PP) per capita, human population divided by area of the country (HP/A), and countries which were treated as dummy variables (independent binomial variables). Assumptions of regression models were evaluated using Durbin–Watson statistics which detect autocorrelation of regression residuals. Pearson's correlation coefficients were used for the evaluation of the relationships between organic agriculture indices and socio-economic variables. Multivariate relationships and multivariate characteristics of the countries based on all the studied variables were evaluated using principal component analysis (PCA) [22]. Kaiser–Meyer–Olkin (KMO) criterion overall and for each variable (MSA—measure of sampling adequacy) as well Bartlett's test of sphericity were applied for the set of studied

variables. Groups of similar countries according to all the studied variables were distinguished using cluster analysis. The analysis was performed using standardized variables and Ward's method for agglomeration based on the squared Euclidean distance between the objects. The statistical analyses were performed using Statistica 13 software [23] and R 4.2.2 package [24]. The significance level for all the analyses was set at 0.05.

Table 2 provides an overview of essential information related to organic agriculture in the EU countries for the year 2021, including the total land area within each EU country designated for organic agriculture practices, the proportion of the country's agricultural land devoted to organic farming, and the extent to which organic agriculture occupied the total land area of each EU country.

**Table 2.** List of EU countries with an area dedicated to organic agriculture, its percentage share in the total utilized agricultural area (UAA), and its percentage share in the country's total area in 2021.

| EU Country | Abbreviation of the Country | The Area under Organic Agric. (1000 ha) | Share of UAA (%) | Share of Total Country Area (%) |
|---|---|---|---|---|
| Austria | AT | 680.8 | 26.2% | 8.1% |
| Belgium | BE | 101.8 | 7.5% | 3.3% |
| Bulgaria | BG | 86.3 | 1.7% | 0.8% |
| Croatia | HR | 121.9 | 7.5% | 1.4% |
| Cyprus | CY | 7.7 | 6.3% | 0.8% |
| Czechia | CZ | 558.0 | 15.8% | 7.1% |
| Denmark | DK | 308.0 | 11.8% | 7.2% |
| Estonia | EE | 226.6 | 23.0% | 5.0% |
| Finland | FI | 365.4 | 16.1% | 1.1% |
| France | FR | 2776.0 | 9.7% | 5.1% |
| Germany | DE | 1582.0 | 8.7% | 4.4% |
| Greece | EL | 534.6 | 9.1% | 4.1% |
| Hungary | HU | 293.6 | 5.8% | 3.2% |
| Ireland | IE | 86.9 | 2.0% | 1.2% |
| Italy | IT | 2187.0 | 17.6% | 7.2% |
| Latvia | LV | 302.2 | 15.3% | 4.7% |
| Lithuania | LT | 261.8 | 8.9% | 4.0% |
| Luxembourg | LU | 6.9 | 5.2% | 2.7% |
| Malta | MT | 0.1 | 0.6% | 0.2% |
| Netherlands | NL | 76.4 | 4.2% | 1.8% |
| Poland | PL | 549.4 | 3.7% | 1.8% |
| Portugal | PT | 308.3 | 6.0% | 3.3% |
| Romania | RO | 578.7 | 4.4% | 2.4% |
| Slovakia | SK | 249.7 | 13.3% | 5.1% |
| Slovenia | SI | 51.8 | 8.4% | 2.5% |
| Spain | ES | 2799.2 | 10.7% | 5.5% |
| Sweden | SE | 606.7 | 20.2% | 1.1% |

## 3. Results

### 3.1. Changes over Time

Table 2 presents linear regression equations analyzing the evolution of the area allocated to organic agriculture over time in the EU countries, along with the coefficients of determination ($R^2$) indicating the goodness of fit of the linear regression model to the data. It should be noticed that for certain countries, strong autocorrelation was observed on the Durbin–Watson statistics for subsequent years, which may have biased the linear regression results. Particularly low values of the Durbin–Watson statistics, below 0.3, were observed for Poland, Denmark, France, and the Netherlands. The changes in organic

farming area over subsequent years were challenging to describe using typical regression functions. Therefore, we opted for the simplest regression model, which was applicable to most countries and facilitated comparison based on average yearly changes (regression coefficients). Figure 2 shows the trends in the area allocated to organic agriculture over time for the three countries with the largest area dedicated to organic agriculture.

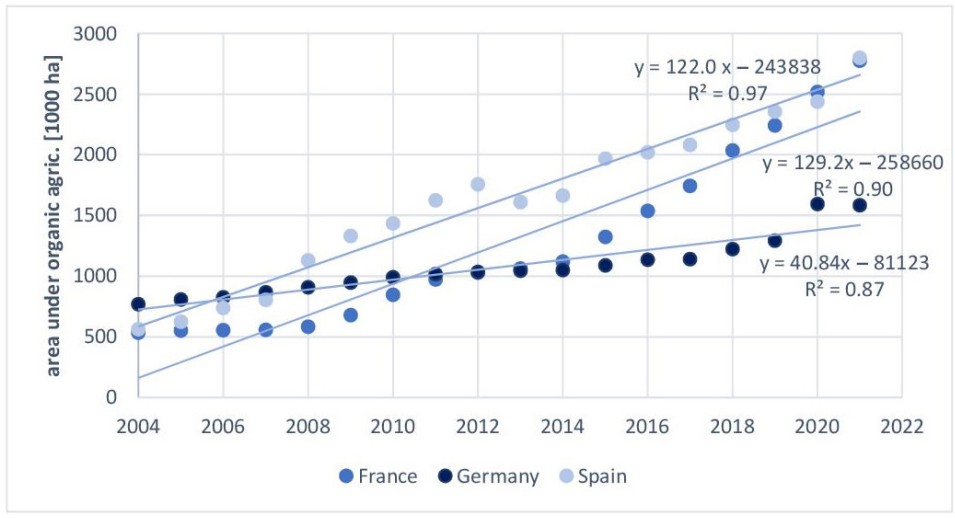

**Figure 2.** Changes in the area dedicated to organic agriculture in France, Germany, and Spain in years from 2004 to 2021.

The linear regression provided insights into the rate of change in the area dedicated to organic agriculture over time. For example, in Austria on average, the area dedicated to organic agriculture increased by approximately 12.9 thousand hectares per year. With an $R^2$ of 0.89 for Austria, the linear model effectively explained 89% of the variation in the changes in the organic agriculture area in Austria. France had the highest coefficient of regression, indicating the fastest growth rate in the area dedicated to organic agriculture per year among all the countries in the table. Spain also showed a substantial growth rate, nearly as high as France. Malta had the lowest coefficient of regression indicating the slowest growth rate in the area dedicated to organic agriculture per year among all the countries in the table. Although higher than Malta, the Netherlands still had a relatively low growth rate compared to the other countries in Table 3. All the values of the coefficient of regression were positive, which meant that the trend over the studied period was positive, i.e., an increase in the area under organic agriculture was observed for all the countries. Table 4 presents the area of organic agriculture as a percentage of the total farmland at the beginning of the study period (2004) and the end of the study period (2021), along with the change presented as percentage points. The highest increase in organic area share was observed in Estonia (17.0%), Sweden (13.1%), and Latvia (12.4%). In Austria, in 2021, the share of organic agriculture was the highest (26.2%) among all the EU countries and increased by 8.8% since 2004. The smallest increase in organic area share during the studied period was observed for Malta (0.5%), Ireland (1.3%), and Portugal (1.5%). The mean organic area share across the EU countries in 2004 was 3.6%, while in 2021, it was 10.0%. The mean change during the study period was 6.4% (percentage points). Evaluation of the trends over time suggested that in most countries, the development of organic farming would not be sufficient to achieve the European Green Deal target of 25% of organic farming by 2030.

**Table 3.** List of EU countries with the linear regression equations, standard errors of estimation (SE$_{est.}$), $p$-values, and coefficient of determination ($R^2$) for the changes in the area dedicated to organic agriculture (in thousand hectares) over time.

| EU Country | Linear Regression | SE$_{est.}$ | $p$-Values | $R^2$ |
|---|---|---|---|---|
| Austria | 12.9·YEAR + 25 380.5 | 24.4 | <0.001 | 0.89 |
| Belgium | 4.9·YEAR − 9 738.4 | 2.5 | <0.001 | 0.99 |
| Bulgaria | 10.8·YEAR − 21 749.0 | 34.9 | <0.001 | 0.72 |
| Croatia | 7.9·YEAR − 15 865.6 | 10.6 | <0.001 | 0.94 |
| Cyprus | 0.3·YEAR − 674.9 | 0.4 | <0.001 | 0.95 |
| Czechia | 18.6·YEAR − 36 931.9 | 39.4 | <0.001 | 0.87 |
| Denmark | 9.9·YEAR − 19 814.0 | 26.7 | <0.001 | 0.81 |
| Estonia | 11.0·YEAR − 22 032.4 | 6.4 | <0.001 | 0.99 |
| Finland | 11.9·YEAR − 23 790.9 | 22.4 | <0.001 | 0.90 |
| France | 129.2·YEAR − 258 660.3 | 232.6 | <0.001 | 0.90 |
| Germany | 40.8·YEAR − 81 122.5 | 87.7 | <0.001 | 0.87 |
| Greece | 16.8·YEAR − 33 441.9 | 57.6 | <0.001 | 0.72 |
| Hungary | 10.3·YEAR − 20 602.2 | 39.7 | <0.001 | 0.67 |
| Ireland | 3.6·YEAR − 7 187.2 | 11.4 | <0.001 | 0.75 |
| Italy | 74.5·YEAR − 148 436.3 | 155.7 | <0.001 | 0.87 |
| Latvia | 12.7·YEAR − 25 315.3 | 24.5 | <0.001 | 0.89 |
| Lithuania | 11.9·YEAR − 23 816.3 | 13.8 | <0.001 | 0.96 |
| Luxembourg | 0.2·YEAR − 403.7 | 0.5 | <0.001 | 0.84 |
| Malta | 0.003·YEAR − 5.9 | 0.0 | <0.001 | 0.63 |
| Netherlands | 1.4·YEAR − 2 844.0 | 6.2 | <0.001 | 0.62 |
| Poland | 22.5·YEAR − 44 850.5 | 130.6 | 0.002 | 0.47 |
| Portugal | 5.0·YEAR − 9 863.8 | 31.0 | 0.003 | 0.44 |
| Romania | 23.2·YEAR − 46 359.5 | 56.6 | <0.001 | 0.81 |
| Slovakia | 6.6·YEAR − 13 014.8 | 18.2 | <0.001 | 0.84 |
| Slovenia | 1.8·YEAR − 3 552.8 | 1.3 | <0.001 | 0.98 |
| Spain | 122.0·YEAR − 243 838.0 | 116.4 | <0.001 | 0.97 |
| Sweden | 25.3·YEAR − 50 523.1 | 36.4 | <0.001 | 0.94 |

**Table 4.** Percentage of organic farming as a share of UAA (%) and the changes between 2004 and 2021 in EU countries.

| Country | Organic Area Share of UAA (%) | | Change (% 2021–% 2004) |
|---|---|---|---|
| | 2004 | 2021 | |
| Austria | 17.6 | 26.2 | 8.6 |
| Belgium | 1.7 | 7.5 | 5.8 |
| Bulgaria | 0.0 | 1.7 | 1.7 |
| Croatia | 0.2 | 7.5 | 7.4 |
| Cyprus | 0.7 | 6.3 | 5.6 |
| Czech Republic | 6.2 | 15.8 | 9.6 |
| Denmark | 5.8 | 11.8 | 5.9 |
| Estonia | 6.0 | 23.0 | 17.0 |
| Finland | 7.3 | 16.1 | 8.8 |
| France | 1.9 | 9.7 | 7.8 |
| Germany | 4.5 | 8.7 | 4.2 |
| Greece | 3.0 | 9.1 | 6.1 |
| Hungary | 3.1 | 5.8 | 2.8 |
| Ireland | 0.7 | 2.0 | 1.3 |
| Italy | 7.3 | 17.6 | 10.4 |

**Table 4.** *Cont.*

| Country | Organic Area Share of UAA (%) | | Change (% 2021–% 2004) |
| --- | --- | --- | --- |
| | 2004 | 2021 | |
| Latvia | 3.0 | 15.3 | 12.4 |
| Lithuania | 1.5 | 8.9 | 7.4 |
| Luxembourg | 2.5 | 5.2 | 2.7 |
| Malta | 0.1 | 0.6 | 0.5 |
| Netherlands | 2.5 | 4.2 | 1.7 |
| Poland | 0.6 | 3.7 | 3.2 |
| Portugal | 4.6 | 6.0 | 1.5 |
| Romania | 0.5 | 4.4 | 3.9 |
| Slovakia | 2.4 | 13.3 | 10.9 |
| Slovenia | 4.7 | 8.4 | 3.7 |
| Spain | 2.9 | 10.7 | 7.8 |
| Sweden | 7.1 | 20.2 | 13.1 |

*3.2. Relationship between Organic Agriculture and Socio-Economic Variables for the Year 2021*

Correlations between the socio-economic indices and variables characterizing organic agriculture are presented in Table 5. The correlations were calculated using data from 2021 across all the EU countries. Most of the correlations were not significant. The only significant correlations were observed between the HDI (human development index) with RC/UAA (percentage of organic area of root crops), BV/UAA (organic bovine animals/utilized agricultural area), PG/UAA (organic pigs/utilized agricultural area) and PO/UAA (organic poultry/utilized agricultural area). All the correlations were positive (in the range between 0.40 and 048), which indicated that a higher HDI was related to a higher percentage of organic root crops and organic livestock density. Most of the other correlations between HDI and variables which described organic agriculture production were not significant but positive, which meant that organic farming was better developed in countries with a higher HDI.

**Table 5.** Correlation coefficients between organic farming variables and socio-economic indices across the EU countries in 2021 (all the abbreviations are explained in Section 2.1).

| | GDPC | HDI | EAS | HP/A |
| --- | --- | --- | --- | --- |
| OAS | −0.02 | 0.21 | −0.20 | −0.36 |
| OA/UAA | −0.01 | 0.21 | −0.21 | −0.35 |
| OP/UAA | −0.16 | 0.10 | −0.07 | −0.04 |
| CR/UAA | 0.08 | 0.23 | −0.13 | −0.30 |
| PP/UAA | −0.03 | 0.12 | −0.06 | −0.29 |
| FR/UAA | −0.33 | −0.29 | 0.08 | −0.15 |
| GR/UAA | −0.06 | 0.03 | −0.10 | −0.01 |
| OL/UAA | −0.23 | −0.12 | 0.15 | −0.17 |
| RC/UAA | 0.31 | 0.46 * | −0.29 | 0.03 |
| VG/UAA | 0.15 | 0.32 | −0.23 | 0.13 |
| BV/UAA | 0.30 | 0.48 * | −0.35 | −0.16 |
| PG/UAA | 0.30 | 0.40 * | −0.22 | 0.02 |
| PO/UAA | 0.30 | 0.47 * | −0.33 | 0.14 |
| SH/UAA | −0.15 | 0.00 | 0.26 | −0.14 |

* significant correlation at 0.05 significance level.

Correlations with other socio-economic indices such as GDPC (gross domestic product per capita), EAS (share employed in agriculture), and HP/A (human population per country area) were not significant. However, some of them were near to significant, e.g., the

correlation between organic agriculture share with human HP/A was negative (r = −0.36). It indicated that the countries with a higher human population density have a lower share of organic agriculture.

To evaluate the existence of multivariate relationships and justify the use of PCA, the Kaiser–Meyer–Olkin (KMO) criterion was calculated for the overall dataset and each variable (MSA—measure of sampling adequacy) in a correlation matrix. Additionally, Bartlett's test of sphericity was conducted for the studied variables. The KMO statistics for overall dataset was 0.503, indicating suitability for PCA, while most variables had MSA statistics above 0.5. The chi-squared value for Bartlett's test of sphericity was 314.3 (df = 136, *p*-value < 0.001), indicating significant correlations between the variables. Therefore, multivariate analysis using PCA was appropriate for evaluating these relationships.

Multivariate relationships between all the studied variables as well as multivariate differences between the studied countries based on the data from 2021 are presented in Figure 3. The strength of the relationship between all the studied variables was moderate as the explanation of total variability by the first two principal components (PC1 and PC2) amounted to 49% (28% for PC1 and 21% for PC2). Most of the variables which characterized organic agriculture were positively correlated; an especially strong positive correlation was observed between PP/UAA (organic area of pulses and protein crops/utilized agricultural area), CR/UAA (organic area of cereals/utilized agricultural area), OAS (organic area share of UAA), RC/UAA (organic area of root crops/utilized agricultural area), and BV/UAA (bovine animals/utilized agricultural area). These six variables showed the strongest positive correlation with PC1, and the highest values of all the six variables were observed for Sweden, Estonia, Denmark, Italy, and Austria. A negative correlation with these six variables was observed with HP/A (human population/area of total country), indicating that these variables were negatively correlated with the human population density. The highest population density as well as the lowest values of the six variables (PP/UAA, OA/UAA, CR/UAA, OAS, RC/UAA, and BV/UAA), were observed in the following countries: Malta, Ireland, and Luxemburg.

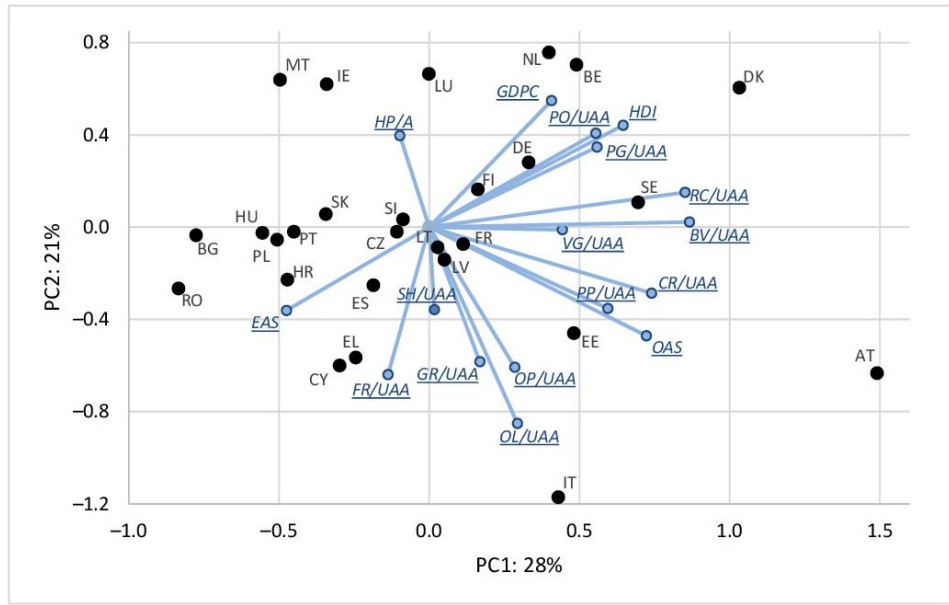

**Figure 3.** Results of PCA presenting multivariate characteristics of the EU countries and relationships between the studied variables that characterize organic agriculture and socio-economic conditions.

Multivariate classification based on all the studied variables is presented in Figure 4. Four groups of countries were distinguished, with Denmark included as an individual country in a separate group. The results were consistent with PCA, for example, AT and IT (Austria and Italy) were included in one group characterized by a very high share of organic

agriculture, organic area of pulses and protein crops/utilized agricultural area, organic area of cereals/utilized agricultural area, organic area of root crops/utilized agricultural area, and bovine animals/utilized agricultural area. Countries located in north-western Europe (DK, NL, and BE) were clustered as one group, characterized by a quite low share of organic area, a low share of people employed in agriculture, and a high density of organic livestock, especially pigs and poultry. These countries have a high human development index. Countries located in Scandinavia, Baltic countries, and Germany (SE, FI, LV, LT, EE, DE) were included in one group which was characterized by high organic agriculture share, which was mainly used for cereal production. These countries had low human population density. Three small countries, MT, LU, and IE, were distinguished as one group which was characterized by a very low organic agriculture share, a very high human population density, and a low share of employed in agriculture. The results of the multivariate analyses revealed significant differences between the EU countries in terms of organic farming development, as well as weak correlations between organic farming and socioeconomic indices.

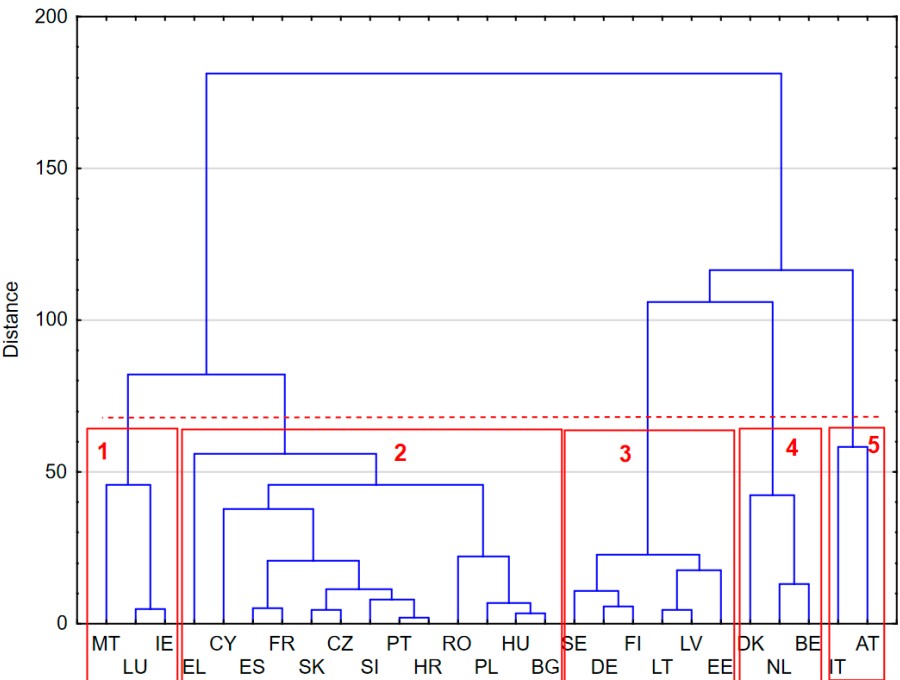

**Figure 4.** Dendrogram based on cluster analysis using the 18 variables for 2021 where groups of EU countries were distinguished. Red rectangles indicate groups of countries similar according to studied variables.

*3.3. Relationship between Organic Agriculture and Socio-Economic Variables for Years 2014–2021 Based on Panel Analysis*

The results of the panel analysis based on multiple linear regression where organic area share of utilized agricultural area (OAS) was the dependent variable and gross domestic product per capita (GDPS), human population per country area, and countries were used as dummy binomial variables are presented in Table 6. The Durbin–Watson statistics were used to evaluate the autocorrelation of residuals in the multiple regression model underlying the panel analysis. The Durbin–Watson value was relatively low at 0.41, indicating some degree of autocorrelation in the residuals, though not exceptionally strong. The results were based on yearly data for the period 2014–2021. Coefficients of regression (0.19) proved the significant positive relationship between GDPC and OAS. It indicated that an increase of GDPC by USD 1000 was related to an increase of OAS by about 0.2%. It meant that higher OAS was usually in the countries with a higher GDPC as well increasing in particular countries along with the increase in GDPC. Human population density had a

negative effect on the OAS. An increase of HP/A by 1 (person/hectare) was related to an increase of the OAS by about 1.8%. It meant that in the countries with a higher population density, the OAS was lower.

**Table 6.** Results of the panel analysis based on multiple regression for organic area share of utilized agricultural area (OAS) as a dependent variable and correlation coefficient between GDPC (in thousand USD) and HP/A with OAS (all the abbreviations are explained in Sections 2.1 and 2.2—Table 1).

| | **Results of Panel Analysis** | | **Correlation Coefficients** | | | |
|---|---|---|---|---|---|---|
| | **Regression Coefficient** | *p*-Value | **with GDPC** | *p*-Value | **with HP/A** | *p*-Value |
| GDPC | 0.19 | <0.001 | | | | |
| HP/A | −1.79 | <0.001 | | | | |
| AT | 13.09 | <0.001 | 0.96 | <0.001 | 0.97 | <0.001 |
| BE | 2.48 | 0.067 | 0.98 | <0.001 | 1.00 | <0.001 |
| BG | −0.80 | 0.117 | 0.82 | <0.001 | −0.84 | <0.001 |
| HR | −0.15 | 0.775 | 0.80 | <0.001 | 0.97 | <0.001 |
| CY | −1.14 | 0.083 | 0.86 | <0.001 | 0.98 | <0.001 |
| CZ | 8.39 | <0.001 | 0.88 | <0.001 | 0.88 | <0.001 |
| DK | 0.57 | 0.433 | 0.94 | <0.001 | 0.92 | <0.001 |
| EE | 9.91 | <0.001 | 0.97 | <0.001 | −0.73 | 0.001 |
| FI | 1.62 | 0.008 | 0.41 | 0.093 | −0.94 | <0.001 |
| FR | −1.22 | 0.064 | 0.89 | <0.001 | 0.87 | <0.001 |
| DE | 1.56 | 0.098 | 0.95 | <0.001 | 0.89 | <0.001 |
| EL | 1.65 | 0.004 | 0.97 | <0.001 | 0.87 | <0.001 |
| HU | 0.12 | 0.841 | 0.95 | <0.001 | −0.99 | <0.001 |
| IE | −9.17 | <0.001 | 0.89 | <0.001 | −0.81 | <0.001 |
| IT | 7.02 | <0.001 | 0.85 | <0.001 | 0.83 | <0.001 |
| LV | 6.86 | <0.001 | 0.93 | <0.001 | 0.38 | 0.122 |
| LT | 1.49 | 0.004 | 0.95 | <0.001 | −0.98 | <0.001 |
| LU | −12.07 | <0.001 | 0.91 | <0.001 | 0.93 | <0.001 |
| MT | 18.78 | <0.001 | 0.93 | <0.001 | −0.94 | <0.001 |
| NL | 0.62 | 0.678 | 0.81 | <0.001 | 0.84 | <0.001 |
| PL | 0.59 | 0.349 | 0.89 | <0.001 | 0.85 | <0.001 |
| PT | 1.95 | 0.002 | 0.65 | 0.004 | 0.00 | 1.000 |
| RO | −0.71 | 0.195 | 0.02 | 0.940 | −0.04 | 0.878 |
| SK | 5.32 | <0.001 | 0.93 | <0.001 | −0.89 | <0.001 |
| SI | 1.92 | 0.002 | 0.94 | <0.001 | 0.94 | <0.001 |
| ES | 1.04 | 0.086 | 0.93 | <0.001 | 0.89 | <0.001 |
| SE | 6.18 | <0.001 | 0.95 | <0.001 | 0.86 | <0.001 |

Correlation coefficients calculated separately for each country proved strong positive correlations between the GDPC with OAS for most of the countries. The exceptions were Finland and Romania, where these correlations were not significant. Positive correlations indicated that the increase in the OAS was related to the increase in the GDPC. These trends were observed in subsequent years, i.e., an increase in the OAS and GDPC were observed simultaneously in subsequent years in the studied period in almost all the countries of the EU. Correlations between the OAS and HP/A were not consistent across the studied countries. Almost all of them were significant; the exceptions were Romania, Latvia, and Portugal. However, for part of the countries, these correlations were positive and for others part of them the correlations were negative. Negative correlations were observed for the countries in which the human population decreased in the studied period and positive correlations were observed for the countries in which the population was increasing. It meant that relationships between the GDPC and HP/A with the OAS were not causal relationships but only interdependencies that existed because of the similar changes in

the studied period. The results of the panel analysis demonstrated significant differences between countries, with a positive effect of GDP per capita (GDPC) on the share of organic farming, as well as a significant negative effect of human population density.

### 3.4. Prediction of Organic Agriculture Share in EU Countries in 2030

Based on the simple regression models presented in Table 2 for the years 2004–2021, the prediction of organic agriculture share in the utilized agricultural area was performed. The results are presented in Table 7. For all the EU countries, the predicted OAS was about 12.1% which was more than two times lower in comparison to the target 25% OAS in the European Commission policy. Very high differences were observed for different EU countries. The target 25% OAS is very probable to be obtained in 2030 in the following countries: Austria, Estonia, and Sweden. In these countries, the predicted OAS in 2030 was about 30% or more. The countries that will probably be near to the target OAS in 2030, i.e., in which the predicted OAS was greater than 20% are as follows: Czechia, Italy, and Latvia. The predicted OAS in 2030 in the other countries was much lower than in most of the EU countries. The lowest predicted OAS, below 5%, was found for Ireland, Netherlands, and Malta.

**Table 7.** Prediction of organic agriculture share in utilized agricultural area in 2030 based on extrapolation using regression functions presented in Table 2.

| Country | Agriculture Area Under Organic Agric. (1000 ha) in 2030 | Utilized Agricultural Area (1000 ha) 2021 | Share in 2030 per Agricultural Area in 2021 |
| --- | --- | --- | --- |
| Austria | 781.3 | 2597.5 | 30.1% |
| Belgium | 146.2 | 1365.7 | 10.7% |
| Bulgaria | 257.9 | 5046.6 | 5.1% |
| Croatia | 190.0 | 1618.0 | 11.7% |
| Cyprus | 10.0 | 123.6 | 8.1% |
| Czechia | 758.0 | 3529.8 | 21.5% |
| Denmark | 369.1 | 2618.0 | 14.1% |
| Estonia | 334.6 | 987.0 | 33.9% |
| Finland | 425.4 | 2268.0 | 18.8% |
| France | 3518.7 | 28,553.8 | 12.3% |
| Germany | 1787.2 | 18,240.0 | 9.8% |
| Greece | 665.2 | 5874.4 | 11.3% |
| Hungary | 347.5 | 5049.2 | 6.9% |
| Ireland | 122.8 | 4337.0 | 2.8% |
| Italy | 2744.5 | 12,403.0 | 22.1% |
| Latvia | 424.4 | 1970.0 | 21.5% |
| Lithuania | 375.5 | 2937.8 | 12.8% |
| Luxembourg | 8.0 | 132.8 | 6.0% |
| Malta | 0.1 | 10.8 | 0.7% |
| Netherlands | 79.0 | 1812.0 | 4.4% |
| Poland | 853.6 | 14,719.5 | 5.8% |
| Portugal | 311.5 | 5121.4 | 6.1% |
| Romania | 654.4 | 13,079.0 | 5.0% |
| Slovakia | 283.4 | 1878.0 | 15.1% |
| Slovenia | 68.4 | 615.3 | 11.1% |
| Spain | 3755.7 | 26,228.4 | 14.3% |
| Sweden | 898.3 | 3002.9 | 29.9% |
| EU | 20,170.7 | 166,119.4 | 12.1% |

One of the main aims of this study was to predict the share of organic farming in 2030. The results indicated that achieving the European Green Deal target of 25% organic

farming by 2030 across the entire EU is not possible, but may be attainable in only a few EU countries.

## 4. Discussion

In all the EU countries during the studied period, an increase in organic agriculture area was observed. The increase ranged from 0.5% for Malta to 17% in Estonia. In most of the countries, the increase was at least several percent points. These changes were consistent with the EU Green Deal policy, which supports the development of organic agriculture [11,25]. The percentage share of organic agriculture in EU countries is larger in comparison to other countries of the world, however, the highest area of organic agriculture is in Australia, with more than 35 million ha, which is over two times higher than in the EU [26].

The report of Willer et al. [26] proved that the significance of organic agriculture in the EU, expressed as the value of annual consumption of organic products per capita, is the highest in EU countries such as Denmark (EUR 344) and Luxemburg (EUR 265), while for the total world, it is several dozen times less (EUR 14). The significance of organic agriculture is increasing in the long term. The European Green Deal includes a target of increasing the total EU agricultural land under organic farming to at least 25% by 2030, which is about three times higher than in 2021. The predictions in this study proved that achieving this goal for the total EU is improbable because the predicted organic agriculture share in 2030 is only about 12% which is more the two times below the expected target. Only three countries, Austria, Estonia, and Sweden, predicted the organic agriculture share which was greater than the expected target. Achieving the goal of a 25% share of organic farming in the EU by 2030 is unrealistic given the current agricultural policy of the EU. This applies not only to organic farming but also to other targets of the European Green Deal that are very difficult or impossible to achieve [27]. Therefore, the targets of the Green Deal should be reformulated, taking into account the feasibility of their implementation. In the case of the development of organic farming in the EU, one of the main limitations could be the large yield gap between conventional and organic farming [28].

Determining the trends of changes and their determinants is therefore important to achieve the assumed goals, i.e., increasing the share of organic agriculture in EU countries. The adoption of organic agriculture depends on many different factors, including the positive effect of organic products on human health [29]. In organic crop production, the use of pesticides is restricted, and antibiotic use is less intensive in organic livestock production. Consumers in wealthier societies are more prone to buying organic agriculture products [30]. Geographical differences are visible between EU countries, with countries that have a higher GDP per capita usually having higher expenditures on organic food products. In almost all the EU member states, a continuous increase of the GDP per capita was observed. The only exception in recent years was Greece where a substantial decrease in the GDP per capita was observed. It is a positive determinant of organic agriculture development because it will increase the need to consume organic products, as well as the area of organic agriculture in EU countries. A study, conducted at the level of smaller administrative units in Croatia, revealed relatively weak correlations between various socio-economic indices and the share of organic agriculture [31]. The strongest positive correlation was observed with the unemployment rate. However, as this socio-economic variable can fluctuate rapidly in time, it is challenging to establish it as a predictor of organic farming development.

Long-term changes in the EU countries indicate an increase in the number and area of organic farms. This is caused by greater consumption of organic products [32] as well as increasing subsidies from the EU for sustainable production, including organic agriculture [33]. Our study has shown a different pattern of development and the current status of organic agriculture across the EU countries.

For example, all three Baltic countries and Austria were characterized by the highest organic area per capita and a large increase in that area during the period of the study

(2004–2021). Another group consisted of Southern European countries (Italy, Spain, France, and Greece), which were characterized by moderate organic area per capita but had a high area of organic fruits and vegetables per capita. In these countries, the main driver of organic agriculture production is the short supply chain, which is much more common in comparison to other countries of the EU [34–36]. A study conducted in Greece [36] proved that one of the main factors correlated with the development of short food supply chains (SFSCs) is the size of the farm, with farmers with smaller cultivation sizes increasing the likelihood of SFSC adoption. SFCS is a tool that can incentivize organic production and provide environmental, economic, and social benefits [37]. The short food chain, which contributes to environmental goals, is attracting more attention in the EU and national laws [38].

Different EU member states have different regulations applied to SFCS, including the definition of "local market", which is connected with the direct selling of organic products in a regional area. A study on determinants of adopting organic farm practices in the Czech Republic [39] proved that positive determinants are subsidies that support organic agriculture, lower farmer age, women as farmers, and smaller size of the farm. A positive highly significant effect of lower age of farmers and smaller size of the farm was proven in another study conducted in Spain [40]. Another study conducted in Latvia and Estonia [41] showed differences between determinants of organic agriculture. Farmers in Latvia were highly responsive to increases in subsidies, whereas farmers in Estonia were relatively unresponsive but more influenced by social factors.

A systemic review that studied factors influencing the adoption of sustainable farming practices in Europe, including organic agriculture, [42] proved the significant effect of farmers' age and education on organic farming adoption. Some studies have evaluated relationships between the development of organic farming with socioeconomic variables in Europe. One of these studies is Stanimir's study conducted for EU countries [43]. In the study, aggregated indicators for organic agriculture were not positively correlated with aggregated factors describing the quality of life and the economic situation of EU countries. The significance of organic agriculture is often higher in countries with lower quality of life.

One of the main limitations in the development of organic agriculture in the EU is economic constraints, caused by lower yields in organic farming in comparison to conventional farming. A meta-analysis conducted by de Ponti et al. [44] showed that currently, organic yields of crops are, on average, around 80% of conventional yields. The difference depends on crop type and the intensity of crop management. The highest yield gap is observed in high-input crop management systems where conventional yields are high; this is especially visible for crops such as wheat and potato. As a result, in countries with less favorable agronomic conditions, especially poor soils, the yield gap between organic and conventional agriculture tends to be lower [45]. While the yield gap between organic and conventional agriculture is quite consistent across different climate conditions, it is slightly higher in warm, moderate climates [46].

Despite differences in organic agriculture development between the EU countries, there is high spatial variability within the countries, and between regions [47–49]. Determinants of organic agriculture development include the significance of subsidies for farmers, which are usually more important in regions where the intensity of agricultural production is lower and farm size is smaller. Another important factor in the development of organic farming is the presence of a long organic heritage, which is associated with socio-cultural factors that lead to different concentrations of organic farming at the regional level.

While our study was conducted at the country level, it was visible that the pattern of organic agriculture development in EU countries was not one-dimensional. Organic agriculture has seen significant growth in Northern Europe (Sweden and Baltic countries) and Central–Southern Europe (Czech Republic, Austria, and Italy). A lower human population density is related to a higher share of organic agriculture area. Higher GDP per capita and HDI are factors that are negatively correlated with organic agriculture areas. However, these correlations were not very strong, indicating that the development of organic farm-

ing depended on many various factors. One of the factors, apart from subsidies, which support the development of organic farming, is farmer education. This includes increasing awareness of the beneficial impact of organic farming on the environment and health, as well as showing the positive economic sides of organic farming [50–52].

Financial support is crucial for development of organic farming in EU countries [53,54]. Currently, support for organic farming is integrated into ecoschemes, which serve as a tool to support agroecological activities at the farm level. Higher subsidies for organic farms, along with typically higher prices for their products, should encourage greater adoption of organic farming. However, it is important to tailor various support tools to the specificities of agriculture in individual countries. This could facilitate the faster development of organic agriculture, particularly in countries where its current share is relatively small.

Increasing the area of organic farming in EU countries, therefore, requires a multi-directional approach, taking into account the attitudes of farmers and society, to ensure greater organic production and a greater demand for organic products. It is also necessary to consider the significant variability in agriculture and socioeconomic factors among EU countries and to adapt agricultural policies to the specific circumstances of each country to facilitate the development of organic farming. This is particularly important given recent political events in Europe, especially those related to the war in Ukraine, which have direct and indirect impacts on agriculture in the EU [55]. Ensuring food security in European countries may become more important than achieving the goals of the Green Deal, potentially leading to slower development of organic farming in the EU.

## 5. Conclusions

The significance of organic agriculture increased in all the EU countries during the study period, especially for countries located in North Europe and Central–South Europe. The development of organic farming was positively correlated with lower human population density. Most of the indices used to characterize the intensity of organic agriculture production were negatively correlated with GDP per capita and HDI. These socioeconomic indices were not very strongly correlated with the development of organic farming.

The multivariate analyses allowed us to distinguish clusters of similar countries according to organic farming production and socioeconomic factors. In the group of the countries with the highest organic farming production were Latvia, Lithuania, Estonia, and Austria, while quite the opposite pattern was observed for Malta, Netherlands, Belgium, Ireland, and Luxemburg.

Overall, while there are challenges and considerations to address, the feasibility of the European Commission's initiative to elevate the share of organic agriculture to 25% by 2030 is only possible with strategic planning, investment in research and development, supportive policies, and collaboration between stakeholders across the agriculture sector. The lack of comprehensive support for the further development of organic agriculture will prevent the EU from achieving the target share by 2030. The current predictions presented in this study indicate that this goal is very unrealistic and almost certainly unattainable. Therefore, it is necessary to reformulate the objectives of the European Green Deal or provide significantly greater financial support for its implementation.

The common Agricultural Policy should prioritize the uneven development of organic farming across different EU countries. Special support for the development of organic farming should be implemented in countries with a low share of organic farming and minimal increases in organic farming area over the study period. Therefore, the common agricultural policy regarding support for organic farming must consider the unique conditions of each EU country.

Given the significant diversity of regions within countries in terms of organic agriculture production, further studies can be performed based on regional data, such as NUTS 1 regions.

It is important to note that the analyses in this study were conducted for the period prior to Russia's invasion of Ukraine, which has impacted agriculture in the EU. As a result,

one of the most significant limitations could be the changes in trends of organic farming development caused by political shifts, as well as recent farmers' protests across the EU against the implementation of the European Green Deal.

**Author Contributions:** Conceptualization, S.K., J.Ž., D.G. and E.W.-G.; methodology, S.K., J.Ž., D.G. and E.W.-G.; validation, D.G. and E.W.-G.; formal analysis, D.G. and E.W.-G.; investigation, S.K., D.G. and E.W.-G.; resources, J.Ž.; data curation, S.K., D.G. and E.W.-G.; writing—original draft preparation, S.K., J. Ž., D.G. and E.W.-G.; writing—review and editing, S.K., J.Ž., D.G., M.C. and E.W.-G.; visualization, S.K., D.G. and E.W.-G.; supervision, E.W.-G.; project administration, J.Ž. and E.W.-G.; funding acquisition, S.K. and J.Ž. All authors have read and agreed to the published version of the manuscript.

**Funding:** This research received no external funding.

**Institutional Review Board Statement:** Not applicable.

**Data Availability Statement:** Data are available upon request from the authors.

**Conflicts of Interest:** The authors declare no conflicts of interest.

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
