# Peer review of "Evaluating the Path to the European Commission’s Organic Agriculture Goal: A Multivariate Analysis of Changes in EU Countries (2004–2021) and Socio-Economic Relationships"

_agriculture, doi:10.3390/agriculture14030477_

Round 1

Reviewer 1 Report (Previous Reviewer 1)

Comments and Suggestions for Authors

The paper still presents many problems, both theoretical and statistical. The work lack of a theoretically sounded research question and has a descriptive nature. From a statistical point of view there are many weakness elements. Many correlations are not significant. In these cases authors cannot comment the sign of the link (Row 215- 217 and Row 220-223). The low correlation indexes between the studied variables suggest that PCA may not be the appropriate tool to synthesise data. Tests such as KMO and Bartlett's test are not presented in the work. They could help to understand whether PCA results are relevant.  Without these tests and with low level of (or not significant) correlations, the results are questionable. 

Author Response

Reviewer 2 Report (Previous Reviewer 2)

Comments and Suggestions for Authors

The manuscript entitled “Evaluating the Path to the European Commission's Organic Agriculture Goal: A Multivariate Analysis of Changes in EU Countries (2004-2021) and Socio-Economic Relationships” investigates shifts in dedicated agricultural areas influenced by evolving preferences and priorities of farmers and consumers. The manuscript adopts a multivariate approach, as well as regression analysis and panel analysis.

In the current version of the manuscript, it still suffers from some major drawbacks, and in particular they refer to used methods. In my opinion, the author(s) use(s) too many methods in just one single manuscript and I still do not get the links and the nexuses among them. For example:

1.       the output of the cluster analysis is not fully exploited in the rest of the manuscript, as further analyses are still done at Member State level

2.       I am skeptical about the analysis in Section 3.1, which is probably mostly biased. Why do(es) the author(s) use linear regression on time-series data, country by country? Among the assumptions of linear regression, “no auto-correlation” is one of them, and in my opinion it is obviously not met here (given the observed trends). However, no checks on that is performed by the author(s)

3.       Rather, I would only keep the panel analysis of section 3.3. and 3.4, provided that some diagnostics tests are performed (e.g. normality test, multicollinearity test, heteroskedasticity test), especially given the little number of observations used.

Moreover, additional limitations refer to:

1.       Abstract. It must be revised, as in its current form it does not reflect the content of the manuscript, and in particular there are no details on the adopted methods.

2.       Introduction: I would put much more emphasis on the EU Strategies (Green Deal, Farm to Fork and Biodiversity) as they are central for the analysis. Conversely, in the current form, they are mentioned only at lines 72-82.

3.       Line 142: the author(s) mention(s) HP as a covariate, while in table 1 is HP/A. Please explain better which variables are used, and assure coherence across the manuscript.

Author Response

Reviewer 3 Report (New Reviewer)

Comments and Suggestions for Authors

The article Evaluating the Path to the European Commission’s organic Agriculture Goal: A Multivariate Analysis of Changes in EU Countries (2004-2021) and Socio-Economic Relationships analyses the landscape dynamic of agricultural areas across Europe by examining the impact  of socio-economic factors and their multivariate relationships across countries in between 2004-2021 period while specifically focusing on areas dedicated to organic agriculture.

The paper convincingly studies changes over time in the evolution of areas dedicated to organic agriculture, and the coherence of the paper and its logic flow is generally well supported by arguments and also in discussions and conclusions. However certain gaps are still to be answered in view of further paper improvements.

-        Besides its general aim the paper should propose certain concrete objectives, research questions or hypotheses. They could be derived and/or be related to relevant existing studies or from the existing context. A literature review chapter would be therefore very useful to introduce these main / reference ideas for the present study.

-        Once established the scientific and the context gap that the paper would like to answer the paper could also better underline its aimed impact and usefulness for possible stakeholders and for scientific literature that treats this topic. This could also be reiterated at the end of discussion or in conclusion chapter as motivating the readability and the usefulness for such a study.

-        Literature on agriculture in Europe using similar methodologies even if different variables or aiming different goals should not be neglected. The analysis of existing studies would confer credibility and would also explain the innovativeness of this study. Please consult and cite some more relevant titles for this study. Some examples  of titles to be consulted by the authors would be: https://doi.org/10.1016/j.jenvman.2011.12.015 - for similar methodologies; https://doi.org/10.1016/j.landusepol.2020.105036 - for predictions explained as difficult to be related to social factors on the topic of organic crops).

-        Besides studies existing on the topic the research might be better contextualized and introduction enlarged as related to this topic and considering the existing EU legislation and policies. Evolutive policy goals to be common but differently translated in the national legislation of the EU countries could also be an important premise for this topic and could be part of the documentation which further may reflect certain results or some discussion issues.

-        Please also explain the utility, the innovativeness and / or the limits of the study in terms of methodology. Why using these indicators to make correlations (some of them not really obviously connected to organic agriculture and very general (e.g. HP/A). Better justifications of indicators should be done (as organic crops refer to quite a specific part of agriculture) and how were they compiled in regression models…which predictors were used in order to forecast the organic agriculture share……  and how these statistic results are sustained in the real context

-        Once the above modifications are brought the discussion and conclusions could be refined and completed accordingly.

  Please take into consideration all the above for further improvements of the paper.

Round 2

Reviewer 1 Report (Previous Reviewer 1)

Comments and Suggestions for Authors

The work is theoretically weak and lack of theoretical explanations of the analysed relationships. Statistical report has been improved with tests but the statistical analysis is not well organized. Moreover, in literature a KMO equal to 0.5 is considered "miserable". 

Author Response

Reviewer 3 Report (New Reviewer)

Comments and Suggestions for Authors

Author Response

This manuscript is a resubmission of an earlier submission. The following is a list of the peer review reports and author responses from that submission.

Round 1

Reviewer 1 Report

Comments and Suggestions for Authors

The work deals with a interesting topic, the changes of organic areas in the EU countries and their relationship with socio-economic variables in last decades. Nevertheless, the article needs to be radically improved. Authors should better specify the questions their analysis want to address and explain which are the theoretical relationship among the variables under analysis by referring to previous studies. As a fact, the choice of socio-economic factors should be better founded. The regression model does not give more informations than an exploratory analysis, as it has not been used to forecast what will be the expected level of organic area in the 2030 and the ability of countries to reach the Green Deal objectives. Perhaps, authors could consider this issue.  Moreover, the discussion and conclusions sections are not well related to the work: they underline the relevance of determinants for the choice of organic agriculture that are not considered in the empirical analysis.

Comments on the Quality of English Language

The quality of the english is good

Reviewer 2 Report

Comments and Suggestions for Authors

The manuscript entitled “Multivariate Characteristics of Organic Farming Changes in EU Countries in 2004-2021 and Relationships with Socio-Economic Variables” analyses the evolution of organic sector in Europe between 2004 and 2021. The manuscript adopts a multivariate approach and addresses the factors that have driven these changes.

In the current version of the manuscript, it seems unsuitable for publication. While the study on the variability in organic farming in the European Union is interesting, it suffers from several concerns that collectively impact the suitability of the paper for publication in our journal. The two primary concerns are:

Methodological Issues

The methodology employed seems a bit chaotic, but insufficiently robust. The authors include too many techniques (linear regression, Pearson’s coefficient, PCA and Cluster analysis) but with no clear explanation of the logic behind the adopted methodologies.

For example, firstly it is not clear which is the role of the selected variables, as they jointly refer to the specific characteristics of the organic sectors and other socioeconomic variables which have nothing to do with it.

Secondly, the application of linear regression appears ambiguous and biased. It is apparent that the authors might be more inclined towards employing panel analysis, especially considering the temporal nature of the analysis. Moreover, I do not understand why other covariates are not added as well.

Thirdly, I am skeptical about the PCA and Cluster analysis grounded on “all the studied variables” (lines 206). What is the justification behind their introduction? Why should also the socioeconomic variables matter in the PCA? In addition, I do not understand the choice of weighting the variables about organic production by human population. (lines 102-117). Why not weighted by UAA? Or by production?

Fourthly, the total absence of references for the adopted methodologies (in particular PCA and CA) and related choices makes even more unclear the reading, as the selected specific methodological choices cannot be appreciated by the reader.

Lastly, referring more in general to the empirical part, it should also be noticed that the statistical analyses at the country level might not adequately capture the complexity of the relationships between socio-economic variables and indices characterizing organic agriculture production, especially in the case of larger countries.

Insufficient Contribution and Limited Discussion

The manuscript has been found lacking in its contribution to the existing body of knowledge on the subject. Firstly, the findings presented in the paper are basically descriptive, and they do not significantly advance the understanding of the factors influencing organic agriculture production in the European Union. Given the availability of this set of variables, the authors could probably favour a panel analysis, leading to more robust findings.

Even the discussion section is not totally satisfactory, as it seems not well connected with the previous analysis. For example, in lines 256-268, consumption of organic products and expenditures are introduced, but they had not been mentioned before. The reader expects that the discussion part is more tightly linked with the previous empirical analysis.

Other minor comments:

The abstract must be totally revised, adding a larger part on the methods (which are absent in the current version) and reducing the description of the results.

Lines 31-47: it is unclear the reference with those topics. Does it fit with organic production?

Lines 60-61: explain more in detail the role of CAP in organic production

Lines 66-67: I expect greater description of the role played by the EU Strategy (i.e. Farm to Fork and Green Deal)

Lines 93-97: there is no need to mention all the EU Member States. Instead, the authors should explain the reason for the exclusion of the UK, which in fact is not mentioned at all.

Lines 97-117: avoid bullet points. Instead provide a table, with proper data description. Moreover,

Table 2: please, provide standard errors, and p.-values, not just R2